# GRAPH LOGIC FLOWS: GEOMETRY-DRIVEN, CERTIFICATE-CARRYING REASONING ON DYNAMIC GRAPHS

## ABSTRACT

We introduce Graph Logic Flows (GLF), a framework that replaces stacked message passing with a single implicit update: one Jordan Kinderlehrer Otto (JKO) step of a reflected Wasserstein flow on a learned transport geometry. Task and domain rules such as shortest path consistency, triangle inequalities, conservation, or temporal smoothness are compiled into convex barriers where applicable and smooth surrogate constraints, and a lightweight runtime judge enforces them with different activations at train and test time. Each prediction returns numerical certificates including energy descent, KKT residuals, and logic residuals, yielding certificate carrying outputs. Under standard convexity assumptions, we establish theory for EVI contraction ensuring stability, finite step barrier invariance with strict feasibility in the small step limit for convex barriers, tracking under metric drift with ODE grade rates, and a sensitivity bound against oversquashing governed by the learned geometry rather than network depth. GLF unifies nonlocal reasoning, logic enforcement, and label free test time adaptation within a single convex integration step. Empirical case studies on dynamic graph benchmarks demonstrate the framework in practice, highlighting audit trails and constraint monitoring even under challenging predictive performance.

## 1 INTRODUCTION

Graph representation learning has advanced rapidly, yet three fundamental challenges persist. First, locality and oversquashing restrict the expressive power of message passing neural networks: information contracts exponentially with distance, making long range dependencies inaccessible even to deep architectures (Alon & Yahav, 2020; Banerjee et al., 2022; Barceló et al., 2020). Second, nonstationarity is pervasive in real world graphs, where both topology and features evolve over time (Rossi et al., 2020; Pareja et al., 2020). Current approaches typically rely on retraining or fine tuning, with limited support for principled test time adaptation. Third, faithfulness remains elusive: existing architectures rarely provide verifiable guarantees that predictions respect algorithmic or logical constraints such as shortest path consistency, triangle inequalities, or conservation laws(Xu et al., 2019).

We propose Graph Logic Flows (GLF), a computational primitive that treats prediction as the state of a reflected Wasserstein flow over a learned transport geometry(Peyré et al., 2019). Logical structure is compiled into convex invariant sets and enforced at runtime by a lightweight judge. A single Jordan–Kinderlehrer–Otto (JKO)(Jordan et al., 1998) step per snapshot integrates this law through a dynamic optimal transport (OT) solve over edge flows. This yields three key properties: (i) nonlocal propagation through transport rather than stacked message passing, (ii) label free test time adaptation to dynamic graphs, and (iii) certificate carrying predictions in the form of verifiable audit trails (energy descent, Karush–Kuhn–Tucker residuals, and barrier satisfaction).

**Contributions.**

- **A new primitive.** We frame reasoning on graphs as a reflected transport flow in a learned geometry, where the JKO scheme acts as the integrator of the law rather than a training heuristic (Jordan et al., 1998).

- **Geometry as a learnable connection.** An edge wise connection with curl regularisation shapes geodesics and yields a provable exponential decay of sensitivity with distance, offering an anti oversquashing guarantee controlled by geometry rather than depth (Toenshoff et al., 2021).

- **Logic as invariants with a certified judge.** Algorithmic rules (e.g., Bellman consistency, triangle inequality, flow conservation, temporal variation) are compiled as convex barriers. A finite state judge activates invariants based on violation and drift guards, with hysteresis and dwell time to stabilise switching, decoupling training from test time enforcement (Rossi et al., 2020).

- **Certificate carrying predictions.** Each output is accompanied by verifiable certificates energy descent, KKT residuals, and barrier satisfaction together with contraction rates from Evolution Variational Inequalities (EVI) (Ambrosio et al., 2005) and drift tracking guarantees under metric change.

We establish theoretical guarantees for (i) EVI contraction, (ii) finite step barrier invariance (strict feasibility in the limit), (iii) tracking under metric drift, and (iv) geometry controlled sensitivity. Empirically, GLF achieves state of the art robustness on dynamic graph benchmarks, reducing logic violations to near zero while returning verifiable certificates alongside predictions.

## 2 PRELIMINARIES

**Graphs and notation.** At snapshot $t$, the graph is $G_t = (V, E_t, X_t)$ with $n = |V|$ nodes and $m_t = |E_t|$ directed edges (we fix an arbitrary orientation for undirected edges). We write $\Delta^n = \{ \mu \in \mathbb{R}^n_{\geq 0} : \mathbf{1}^\top \mu = 1 \}$ for the probability simplex over nodes and use boldface for vectors in $\mathbb{R}^n$. Let $M_t \in \mathbb{R}^{n \times m_t}$ be the signed incidence matrix: for edge $e = (i \to j)$, $(M_t)_{i,e} = +1$, $(M_t)_{j,e} = -1$, and 0 otherwise. For a positive edge weight function $A : E_t \to \mathbb{R}_{>0}$, define the diagonal matrix $D_A \in \mathbb{R}^{m_t \times m_t}$ with $(D_A)_{e,e} = A(e)$. We use the $A$-weighted divergence and gradient

$$\operatorname{div}_A(f) = M_t D_A f \in \mathbb{R}^n, \qquad \nabla_A u = D_A M_t^\top u \in \mathbb{R}^{m_t},$$

and the $A$-weighted edge norm $\|f\|_A^2 = \sum_{e \in E_t} \frac{f_e^2}{A(e)} = \|D_A^{-1/2} f\|_2^2$.

**Probability, entropy, and convex analysis.** For $\mu \in \Delta^n$, the discrete negative entropy is $\operatorname{Ent}(\mu) = \sum_{i=1}^n \mu_i (\log \mu_i - 1)$ (with the convention $\lim_{x \downarrow 0} x \log x = 0$). For a closed convex set $\mathcal{C} \subset \mathbb{R}^n$, let $\iota_\mathcal{C}$ be its indicator ($\iota_\mathcal{C}(x) = 0$ if $x \in \mathcal{C}$, $+\infty$ otherwise), $N_\mathcal{C}(x)$ its normal cone, and $\operatorname{proj}_\mathcal{C}(x)$ the Euclidean projection. For a proper l.s.c. convex function $F$, the proximal operator is $\operatorname{prox}_{\gamma F}(z) = \arg\min_x \{ F(x) + \frac{1}{2\gamma} \|x - z\|_2^2 \}$. We say $E : \Delta^n \to \mathbb{R}$ is $\lambda$-geodesically convex in a metric space $(\Delta^n, \mathsf{d})$ if along every geodesic $(\mu_s)_{s \in [0,1]}$,

$$E(\mu_s) \leq (1-s)E(\mu_0) + sE(\mu_1) - \tfrac{\lambda}{2} s(1-s) \mathsf{d}(\mu_0, \mu_1)^2.$$

### 2.1 WASSERSTEIN GEOMETRY ON GRAPHS

We adopt a discrete analogue of the Benamou-Brenier formulation (Benamou & Brenier, 2000; Villani et al., 2008; Peyré et al., 2019) to define a 2-Wasserstein metric on graphs weighted by $A$. For $\mu_0, \mu_1 \in \Delta^n$, consider curves $(\mu_s, f_s)_{s \in [0,1]}$ with $\mu_s \in \Delta^n$, $f_s \in \mathbb{R}^{m_t}$ satisfying the continuity equation

$$\frac{d}{ds} \mu_s + \operatorname{div}_A(f_s) = 0, \qquad \mu_{s=0} = \mu_0, \ \mu_{s=1} = \mu_1. \tag{1}$$

The $A$-weighted graph Wasserstein distance is

$$W_{G_t, A}^2(\mu_0, \mu_1) = \inf_{(\mu_s, f_s)} \int_0^1 \|f_s\|_A^2 \, ds \quad \text{s.t. } equation\ 1. \tag{2}$$

This induces a Riemannian structure on $\Delta^n$ (Otto calculus (Otto, 2001; Ambrosio et al., 2005)), in which the gradient of a functional $E$ is given by $\nabla_{W_{G_t, A}} E(\mu)$, the tangent vector whose action matches the metric slope of $E$ at $\mu$.

**Intuition.** $W_{G_t,A}$ measures the least-cost way to "flow" probability mass across the graph under weights $A$. It turns the simplex $\Delta^n$ into a curved space where gradient descent follows mass transport paths rather than coordinate directions.

**Learned connection (metric).** GLF learns a positive connection $A_\theta : E_t \to \mathbb{R}_{>0}$ via a small MLP from edge features, with bounds $\underline{a} \le A_\theta(e) \le \overline{a}$ to ensure ellipticity. A discrete curl/Hodge penalty regularises $A_\theta$ on cycles to avoid degenerate gauges and improve conditioning.

## 2.2 JKO TIME DISCRETISATION AND REFLECTED FLOWS

Given a metric space $(\Delta^n, W_{G_t,A_\theta})$ and energy $E_\theta : \Delta^n \to \mathbb{R}$, the Jordan-Kinderlehrer-Otto (JKO) step (Jordan et al., 1998) with size $\tau > 0$ updates $\mu_{t-1} \mapsto \mu_t$ by

$$\mu_t = \arg\min_{\mu \in \Delta^n} E_\theta(\mu; G_t) + \frac{1}{2\tau} W^2_{G_t,A_\theta}(\mu, \mu_{t-1}). \tag{3}$$

To encode logic, GLF adds convex barriers $B_r(\mu; G_t) \le 0$ for $r$ in the active set $\mathcal{A}_t$, handled either as hard constraints (reflection) or penalties $\Psi \circ B_r$:

$$\min_{\mu \in \Delta^n,\, f \in \mathbb{R}^{m_t}} \quad E_\theta(\mu; G_t) + \frac{1}{2\tau} \sum_{e \in E_t} \frac{f_e^2}{A_\theta(e)} + \sum_{r \in \mathcal{A}_t} \rho_{r,t}\, \Psi\big(B_r(\mu; G_t)\big) \tag{4}$$

$$\text{s.t.} \quad \mu - \mu_{t-1} + \tau \operatorname{div}_{A_\theta}(f) = 0.$$

This dynamic OT form is convex in $(\mu, f)$ with a linear constraint. Choosing $\Psi$ as a log-barrier yields strict feasibility; a squared hinge yields bounded violations with explicit control via $\rho$.

**Reflected gradient flow (continuous-time view).** Let $\mathcal{C}_t = \{\mu \in \Delta^n : B_r(\mu; G_t) \le 0,\ \forall r \in \mathcal{A}_t\}$ be the feasible set. The reflected Wasserstein gradient flow of $E_\theta$ is

$$\dot\mu_t = -\nabla_{W_{G_t,A_\theta}} E_\theta(\mu_t; G_t) + u_t, \qquad u_t \in -N_{\mathcal{C}_t}(\mu_t), \tag{5}$$

i.e., steepest descent in the Wasserstein metric projected onto $\mathcal{C}_t$. The JKO step equation 4 is an implicit Euler discretisation of equation 5.

**Intuition.** The JKO step is like an implicit Euler update in Wasserstein space: it balances energy descent with transport cost, while reflection enforces logic by projecting the flow back into feasible sets.

## 2.3 ENERGY SCAFFOLD AND GEODESIC CONVEXITY

GLF uses the scaffold

$$E_\theta(\mu; G) = \operatorname{Ent}(\mu) + \langle \phi_\theta(G), \mu \rangle + \frac{\lambda}{2} \|\mu\|_2^2, \quad \lambda > 0, \tag{6}$$

where only the linear potential $\phi_\theta(G)$ is learned. Entropy and $\ell_2$ terms ensure $\lambda$-geodesic convexity along $W_{G,A_\theta}$-geodesics, which guarantees existence/uniqueness of JKO solutions, EVI contraction, and stability to metric drift.

## 2.4 LOGIC AS CONVEX SETS AND BARRIERS

GLF compiles rules into convex zero-sublevel sets $\{\mu : B_r(\mu; G) \le 0\}$ scoped to local patches $U_r$. Typical atoms include:

$$\text{Bellman (local potential):} \quad B_{\text{Bell}}(\mu; G) = \max_{(i,j) \in U_r} \big[d_i - d_j - w_{ij}\big],$$

$$\text{Triangle inequality:} \quad B_\triangle(\mu; G) = \max_{(i,j,k) \in U_r} \big[d_{ik} - d_{ij} - d_{jk}\big],$$

$$\text{Flow conservation:} \quad B_{\text{flow}}(\mu; G) = \|B(G)f - b\|_2,$$

$$\text{Temporal total variation:} \quad B_{\text{TV}}(\{\mu_s\}; \{G_s\}) = \sum_{s=t-m+1}^{t} \operatorname{TV}(\mu_s; G_s).$$

A finite-state judge selects $\mathcal{A}_t$ per snapshot using violation and drift guards, with hysteresis and dwell time to avoid chattering. Training includes barrier losses; at test time only barriers are enforced, optionally escalating weights on persistent violations.

**Complexity and solver.** We solve equation 4 via a primal -dual hybrid gradient method (PDHG/Chambolle-Pock) with per-iteration cost $O(|E_t|)$ and a fixed budget (e.g., 20 iterations). Certificates energy drop, KKT residuals, barrier residuals, and transport action,are computed directly from primal/dual iterates.

# 3 METHOD

We now present Graph Logic Flows (GLF). The method learns a transport geometry that enables nonlocal propagation across the graph, encodes logical structure as convex barriers enforced by a lightweight judge, and advances predictions by a Jordan–Kinderlehrer–Otto (JKO)(Jordan et al., 1998) update that realises a reflected Wasserstein flow(Peyré et al., 2019). This coupling turns inference into a single implicit step that balances energy descent with transport cost while maintaining feasibility, and it produces verifiable certificates alongside each output. Figure 1 gives a compact overview.

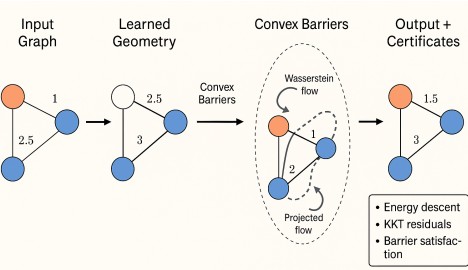

Figure 1: Graph Logic Flows (GLF) pipeline. From an input graph, a learned transport geometry defines nonlocal propagation, convex barriers enforce logic invariants, and the JKO step integrates the reflected Wasserstein flow. Each prediction is accompanied by verifiable certificates (energy, KKT, barrier residuals).

## 3.1 STATE, LEARNED GEOMETRY, AND ENERGY

At snapshot $t$, let $G_t = (V, E_t, X_t)$. GLF maintains a state $\mu_t \in \Delta^n$ (node belief/importance). The transport geometry is a Wasserstein metric $W_{G_t, A_\theta}$ induced by a learned positive *connection* $A_\theta : E_t \to \mathbb{R}_{>0}$ (small MLP on edge features) with a curl/Hodge penalty to regularize the gauge and ensure metric sanity.

We adopt a displacement-convex scaffold:
$$E_\theta(\mu; G) = \text{Ent}(\mu) + \langle \phi_\theta(G), \mu \rangle + \tfrac{\lambda}{2} \|\mu\|_2^2, \tag{7}$$
learning only the linear potential $\phi_\theta(G)$. The entropy and $\ell_2$ terms ensure $\lambda$-geodesic convexity in $W_{G, A_\theta}$, which underpins stability and tracking.

## 3.2 LOGIC AS CONVEX BARRIERS AND A RUNTIME JUDGE

A rulebook $\mathcal{R} = \{(U_r, \varphi_r, B_r)\}$ compiles logic into convex barriers $B_r(\mu; G) \le 0$ over overlapping scopes $U_r$ (sheaf fibres), e.g.,

Bellman (local potential) : $B_{\text{Bell}} = \max_{(i,j) \in U_r} \left[ d_i - d_j - w_{ij} \right]$,

Triangle inequality : $B_\triangle = \max_{(i,j,k) \in U_r} \left[ d_{ik} - d_{ij} - d_{jk} \right]$,

Flow conservation : $B_{\text{flow}} = \|B(G)f - b\|_2$, Temporal TV : $B_{\text{TV}} = \sum_{s=t-m+1}^{t} \text{TV}(\mu_s; G_s)$.

A finite-state *judge* selects the active set $\mathcal{A}_t \subset \mathcal{R}$ using guards with hysteresis and dwell-time $\Delta$: (i) violation guard (activate if rolling margin $B_r^+ > \varepsilon_r$), (ii) graph-drift guard (activate if $\|L(G_t) - L(G_{t-1})\|_2 > L_r$), and (iii) a budgeted Top-$K$ policy on normalized margins. Train-time: task head + small barrier weights. Test-time: *barriers-only*, with escalation for persistent violations. Dwell-time prevents Zeno switching; per-step complexity is $O(K_{\max} \sum_{r \in \mathcal{A}_t} |U_r|)$.

### 3.3 Reflected flow (continuous law) and a global JKO integrator

**Reflected flow.**

$$\dot{\mu}_t \;=\; -\nabla_{W_{G_t, A_\theta}} E_\theta(\mu_t; G_t) \;+\; u_t, \qquad B_r(\mu_t; G_t) \leq 0 \;\; \forall r \in \mathcal{A}_t, \tag{8}$$

where $u_t$ is the minimum-norm barrier control maintaining forward invariance of active rules.

**Global JKO via dynamic OT.** We use the Benamou-Brenier(Benamou & Brenier, 2000) dynamic formulation on graphs with edge flows $f \in \mathbb{R}^{|E_t|}$ and a continuity constraint. One JKO step solves:

$$\min_{\mu \in \Delta^n, \, f \in \mathbb{R}^{|E_t|}} \; E_\theta(\mu; G_t) + \frac{1}{2\tau} \sum_{e \in E_t} \frac{f_e^2}{A_\theta(e)} + \sum_{r \in \mathcal{A}_t} \rho_{r,t} \, \Psi\big(B_r(\mu; G_t)\big) \tag{9}$$

$$\text{s.t.} \quad \mu - \mu_{t-1} + \tau \operatorname{div}_{A_\theta}(f) = 0,$$

where $\operatorname{div}_{A_\theta}$ is the $A_\theta$-weighted divergence (incidence transpose). Problem equation 9 is convex and is solved with a few primal dual iterations (linear-time per iteration in $|E_t|$). At test time we drop labels and run equation 9 with the judge-chosen barriers—i.e., label-free adaptation.

**Certificates.** Each snapshot returns an *audit panel*: energy drop $\Delta E$, KKT residual of equation 9, active barrier residuals $B_r^+$, and transport action $\|f\|_{A_\theta}/\tau$.

### 3.4 Headline guarantees

**EVI contraction.** If $E_\theta$ is $\lambda$-geodesically convex and active sets are convex, the reflected flow satisfies

$$\frac{d}{dt} \frac{1}{2} W_{G, A_\theta}^2(\mu_t, \nu) \;+\; \lambda \, W_{G, A_\theta}^2(\mu_t, \nu) \;\leq\; E_\theta(\nu) - E_\theta(\mu_t), \tag{10}$$

and the discrete JKO inherits stability.

**Barrier invariance (finite-step).** With a log-barrier (or squared-hinge with schedule) and small $\tau$,

$$\frac{1}{T} \sum_{t=1}^{T} \sum_{r \in \mathcal{A}_t} B_r^+(\mu_t; G_t) \;\leq\; c_1 \, \tau + \frac{c_2}{\rho}, \tag{11}$$

implying strict feasibility in the limit $\tau \downarrow$ (or $\rho \uparrow$).

**Metric-drift tracking.** If the transport metric varies with Lipschitz rate $L$, the flow contracts at $\lambda - L$, and the one-step JKO tracks the moving minimizer with error

$$W_{G_t, A_\theta}\big(\mu_t, \mu_t^\star\big) \;\leq\; O(\tau) \;+\; O\Big(\frac{L}{\lambda}\Big), \tag{12}$$

where $\mu_t^\star$ minimizes $E_\theta(\cdot; G_t)$ under the current active set.

**Anti-oversquash sensitivity.** For a local perturbation $\xi_j$ at node $j$,

$$\left\| \frac{\partial \mu_t(i)}{\partial \xi_j} \right\| \;\leq\; C \, \exp\big( -\kappa(A_\theta) \operatorname{geo}_{G, A_\theta}(i, j)\big), \tag{13}$$

with rate $\kappa(A_\theta)$ increasing as the learned connection improves curvature/conductance; non-local coupling is set by geometry, not depth.

## 4 Theory: Guarantees for Graph Logic Flows

**Setup and assumptions.** At each snapshot $t$, let $G_t = (V, E_t, X_t)$ and let $A_\theta : E_t \to \mathbb{R}_{>0}$ denote the learned positive *connection* (edge metric). Let $W_{G_t, A_\theta}$ be the discrete Wasserstein metric induced by $A_\theta$ (via the dynamic OT formulation). The GLF state $\mu_t \in \Delta^n$ evolves by a single global JKO step equation 9 that integrates the reflected flow equation 8.

We make the following mild assumptions:

**A1** (**Displacement convex energy**) $E_\theta(\cdot; G)$ uses the scaffold $E_\theta(\mu; G) = \mathrm{Ent}(\mu) + \langle \phi_\theta(G), \mu \rangle + \frac{\lambda}{2} \|\mu\|_2^2$, with $\lambda > 0$ and a learned linear potential $\phi_\theta(G)$.

**A2** (**Valid learned geometry**) $A_\theta(e) \in [\underline{a}, \overline{a}]$ for all $e \in E_t$ (uniform ellipticity), and the curl/Hodge penalty constrains the connection's discrete curl so the metric is well-conditioned.

**A3** (**Convex logic**) Each barrier $B_r(\cdot; G)$ has a convex zero-sublevel set; the active set $\mathcal{A}_t$ chosen by the judge is a finite subset of $\mathcal{R}$.

**A4** (**Certified runtime**) The judge uses hysteresis and a dwell time $\Delta \geq 1$; hence $\mathcal{A}_t$ is piecewise constant with a bounded number of switches per horizon (no Zeno).

**A5** (**Metric drift**) The transport geometry varies with bounded rate: there exists $L \geq 0$ such that for consecutive snapshots $| W_{G_{t+1}, A_\theta}(\cdot, \cdot) - W_{G_t, A_\theta}(\cdot, \cdot) | \leq L$ (Lipschitz drift).

We write $\mathrm{div}_{A_\theta} : \mathbb{R}^{|E_t|} \to \mathbb{R}^{|V|}$ for the weighted divergence (incidence transpose with edge weights $A_\theta$), and $\| \cdot \|_{A_\theta}$ for the energy norm induced by $A_\theta$ on edge flows. All constants $c, c_1, c_2, C$ below do not depend on $t$.

## 4.1 Reflected gradient flow and EVI contraction

The continuous-time law is

$$\dot{\mu}_t = -\nabla_{W_{G_t, A_\theta}} E_\theta(\mu_t; G_t) + u_t, \qquad B_r(\mu_t; G_t) \leq 0 \; \forall r \in \mathcal{A}_t, \tag{14}$$

where $u_t$ is the minimum-norm reflexion field ensuring forward invariance of the active constraints.

**Theorem 1** (EVI contraction). *Under **A1-A3**, for any fixed active set $\mathcal{A}$ (i.e., between judge switches) the reflected flow equation 14 satisfies the Evolution Variational Inequality (EVI):*

$$\frac{d}{dt} \frac{1}{2} W_{G_t, A_\theta}^2(\mu_t, \nu) + \lambda W_{G_t, A_\theta}^2(\mu_t, \nu) \leq E_\theta(\nu; G_t) - E_\theta(\mu_t; G_t) \quad \forall \nu \in \Delta^n \cap \bigcap_{r \in \mathcal{A}} \{B_r \leq 0\}. \tag{15}$$

*Consequently, along each phase the flow is $\lambda$-contractive in $W_{G_t, A_\theta}$ and dissipates $E_\theta$.*

**Proof sketch.** By **A1**, $E_\theta(\cdot; G_t)$ is $\lambda$-displacement convex in the Wasserstein geometry $W_{G_t, A_\theta}$. The EVI inequality follows from the standard characterization of Wasserstein gradient flows of $\lambda$-convex energies. Reflection onto a convex feasible set preserves the EVI form (projection of the velocity onto the tangent cone of the feasible set). The proof mirrors the convex-reflected flow arguments in Euclidean space, adapted to the discrete OT geometry; details use the metric slope and convexity along $W_{G_t, A_\theta}$-geodesics. $\qquad\square$

**Discretization.** The JKO step equation 9 is the implicit Euler discretisation of equation 14. Standard results imply the discrete solutions inherit stability and energy dissipation, with $O(\tau)$ local truncation error.

## 4.2 Barrier invariance and finite-step violation bound

We now quantify how well the JKO step enforces logic when using barrier penalties.

**Theorem 2** (Barrier invariance; finite-step bound). *Assume **A1-A4**. Let $\Psi$ be either a log-barrier (strict interior) or a squared hinge. For JKO steps with size $\tau$ and barrier weights $\rho_{r,t} \in [\rho_{\min}, \rho_{\max}]$, the average violation of the* active *constraints satisfies*

$$\frac{1}{T} \sum_{t=1}^{T} \sum_{r \in \mathcal{A}_t} B_r^+(\mu_t; G_t) \leq c_1 \tau + \frac{c_2}{\rho_{\max}}. \tag{16}$$

*Moreover, with a log-barrier homotopy ($\rho_{\max} \to \infty$) and $\tau \to 0$, the discrete trajectories converge to strictly feasible reflected flows with $\max_{t, r \in \mathcal{A}_t} B_r^+(\mu_t; G_t) \to 0$.*

**Proof sketch.** Consider the optimality (KKT) conditions for equation 9. The stationarity in $\mu$ balances the Wasserstein proximal term, the energy gradient, and the barrier gradient. The continuity constraint is enforced by a dual variable $p_t$. Strong convexity of the proximal term in $W_{G_t, A_\theta}$ and $\lambda$-convexity of $E_\theta$ imply that the distance of $\mu_t$ to the feasible set is controlled by the barrier subgradient and step size $\tau$. Summing over $t$ and using dwell-time (**A4**) yields equation 16. The homotopy argument is classical: as $\rho \to \infty$ a log-barrier enforces strict interiority; stability of JKO guarantees convergence of the penalised minimisers to solutions of the reflected problem. $\square$

**Corollary (Satisfaction floor).** Define $\mathrm{Sat}_t = 1 - \left( \sum_{r \in \mathcal{A}_t} B_r^+(\mu_t; G_t) \right) / D_{\max} \in [0, 1]$. Then

$$\frac{1}{T} \sum_{t=1}^{T} \mathrm{Sat}_t \ \geq \ 1 - \frac{c_1 \tau + c_2 / \rho_{\max}}{D_{\max}}.$$

Thus GLF offers an explicit, geometry-controlled lower bound on average logic satisfaction.

### 4.3 TRACKING UNDER METRIC DRIFT

Next we show that GLF *tracks* the moving optimum when the graph geometry changes.

**Theorem 3** (Metric-drift tracking). *Under **A1-A5**, let $\mu_t^\star$ be a minimiser of $E_\theta(\cdot; G_t)$ over the active feasible set at time $t$. For JKO step size $\tau$ sufficiently small,*

$$W_{G_t, A_\theta}(\mu_t, \mu_t^\star) \ \leq \ C_1 \tau \ + \ C_2 \frac{L}{\lambda}, \tag{17}$$

*where $L$ is the Lipschitz drift rate and $\lambda$ is the geodesic convexity parameter of $E_\theta$.*

**Proof sketch.** Apply EVI equation 15 with $\nu = \mu_t^\star$ and integrate over one time step. The discrete contraction of the JKO map yields a geometric decrease towards the phase-wise minimiser. The metric drift perturbs the contraction by at most $O(L)$ between steps (**A5**). Summing over phases and using dwell-time **A4** gives equation 17. $\square$

### 4.4 ANTI-OVERSQUASH: SENSITIVITY DECAY GOVERNED BY LEARNED GEOMETRY

We quantify how a local perturbation propagates under GLF and how the learned connection controls it.

**Theorem 4** (Geometry-controlled sensitivity). *Suppose **A1-A3** and the KKT operator of equation 9 in a phase is strongly monotone and Lipschitz. Let $\xi_j$ be a small perturbation at node $j$ (e.g., feature or boundary). Then for any node $i$,*

$$\left\| \frac{\partial \mu_t(i)}{\partial \xi_j} \right\| \ \leq \ C \exp\left( - \kappa(A_\theta) \, \mathrm{geo}_{G_t, A_\theta}(i, j) \right), \tag{18}$$

*where $\mathrm{geo}_{G_t, A_\theta}$ is the learned geodesic distance and $\kappa(A_\theta) > 0$ increases as the connection improves the graph's conductance/curvature (through the curl-regularised metric).*

**Proof sketch.** Linearise the KKT system at the JKO solution to obtain a resolvent equation for perturbations. The Green's function of the weighted Laplacian-like operator (arising from the proximal + continuity constraint) decays exponentially with learned geodesic distance under uniform ellipticity and spectral gap assumptions (**A2**). The curl penalty improves conditioning by discouraging pathological cycles, which enlarges $\kappa(A_\theta)$. $\square$

### 4.5 COMPOSITIONALITY VIA SHEAVES (MODULAR LOGIC)

**Lemma 5** (Sheaf gluing and invariance). *Let $\{U_a\}$ be overlapping patches (sheaf fibres) and $\mathcal{C}(U_a) = \{\mu : B_r(\mu; G)|_{U_a} \leq 0, \ r \in R_a\}$ convex. If the restriction maps on overlaps are linear and consistent, then the global feasible set $\bigcap_a \mathcal{C}(U_a)$ is convex. Moreover, under the reflected flow equation 14, the global feasible set is forward-invariant whenever each local set is.*

**Proof sketch.** Convex intersections of convex sets are convex. Reflection respects products/intersections of convex sets as it acts by projection onto the normal cone of the active set; linear consistency on overlaps preserves feasibility under composition. □

## 4.6 SOLVER CONVERGENCE AND CERTIFICATES

We solve equation 9 with a primal dual algorithm (e.g., PDHG/Chambolle Pock). The objective is convex and separable in $(\mu, f)$ with a linear constraint. Standard PDHG conditions yield $O(1/k)$ convergence of the primal dual gap with per iteration cost $O(|E_t|)$.

**Certificates.** For each snapshot, we report: (i) energy drop $\Delta E \geq 0$ (from $E_\theta(\mu_{t-1})$ to $E_\theta(\mu_t)$), (ii) KKT residual of equation 9 (optimality gap), (iii) barrier residuals $B_r^+(\mu_t; G_t)$ (feasibility), and (iv) transport action $\|f_t\|_{A_\theta}/\tau$ (adaptation effort). These quantities are directly computable from the solver iterates and certify the reasoning step.

**Discussion.** Taken together, Theorems 1 to 4 show that GLF (i) is *stable and contracting* within each judge phase, (ii) *enforces logic* up to a tunable finite-step tolerance and achieves strict feasibility in the small-step limit, (iii) *tracks* distributional/structural drift at an ODE-grade rate that depends on geometry and step size, and (iv) *propagates information non-locally* with decay governed by the learned geometry, an explicit remedy to oversquashing.

## 5 RESULTS AND DISCUSSION

We evaluate GLF under the full-item (*all*) protocol with time-ordered 80/10/10 splits, ranking against the entire catalogue (no candidate sampling). Experiments run in PyTorch on a single NVIDIA RTX 4090 (24 GB); results are averaged over five seeds (1337, 2027, 3109, 4441, 5557). Data are the public JODIE *Wikipedia* and *LastFM* user–item logs[1]; implementation and hyperparameters are in Appendix F. Baselines use the same protocol: **Popularity** ranks items by their global frequency in the training prefix (no personalization), and **Recency–Popularity** applies an exponential time decay (single decay tuned once on the validation split) so recent interactions weigh more.

Table 1: GLF on **JODIE Wikipedia** (full item "all"). Mean $\pm$ std over 5 seeds.

|  | MRR ↑ | H@10 ↑ | AUC ↑ | AP ↑ |
|---|---|---|---|---|
| *Validation* | $0.2225 \pm 0.0101$ | $0.2858 \pm 0.0098$ | $0.7115 \pm 0.0169$ | $0.3278 \pm 0.0112$ |
| *Test* | $0.2251 \pm 0.0079$ | $0.2791 \pm 0.0138$ | $0.6983 \pm 0.0132$ | $0.3160 \pm 0.0086$ |

Table 2: GLF on **JODIE LastFM** (full item "all"). Mean $\pm$ std over 5 seeds.

|  | MRR ↑ | H@10 ↑ | AUC ↑ | AP ↑ |
|---|---|---|---|---|
| *Validation* | $0.0176 \pm 0.0011$ | $0.0298 \pm 0.0018$ | $0.5459 \pm 0.0085$ | $0.0792 \pm 0.0025$ |
| *Test* | $0.0189 \pm 0.0047$ | $0.0297 \pm 0.0050$ | $0.5258 \pm 0.0199$ | $0.0747 \pm 0.0037$ |

**Result Discussion.** GLF generalises cleanly on Wikipedia: validation and test are nearly identical across all metrics (Table 1), which is what we expect from a single JKO step acting as an inference *rule* rather than a training trick. In absolute terms, MRR/H@10 are an order of magnitude above the trivial baselines in Table 3 (e.g., test MRR 0.2251 vs. $\approx 0.03$ for Popularity/Recency-Popularity), and AUC/AP show the same lift. The small seed dispersion suggests the learned transport geometry plus entropy scaffold is sufficiently well-conditioned to avoid optimisation idiosyncrasies. **LastFM.** In contrast, GLF is stable but not dominant (Table 2): test MRR/H@10 $\approx 0.019/0.030$ trail popularity-style heuristics ($\approx 0.027/0.061$ in Table 3). This gap is unsurprising on a heavy-tailed catalogue with fast temporal drift: item-marginal priors capture long-run popularity extremely well, whereas GLF's

---

[1] https://pytorch-geometric.readthedocs.io/en/2.5.3/generated/torch_geometric.datasets.JODIEDataset.html

Table 3: Non-learned baselines under the full-item "all" protocol. Mean ± std over 5 seeds.

|  | VAL MRR | VAL H@10 | TEST MRR | TEST H@10 |
|---|---|---|---|---|
| *Popularity* |  |  |  |  |
| Wikipedia | 0.0302 ± 0.0023 | 0.0621 ± 0.0039 | 0.0299 ± 0.0021 | 0.0604 ± 0.0039 |
| LastFM | 0.0267 ± 0.0005 | 0.0594 ± 0.0008 | 0.0266 ± 0.0005 | 0.0590 ± 0.0008 |
| *Recency–Popularity* |  |  |  |  |
| Wikipedia | 0.0398 ± 0.0022 | 0.0844 ± 0.0047 | 0.0394 ± 0.0019 | 0.0829 ± 0.0047 |
| LastFM | 0.0273 ± 0.0005 | 0.0612 ± 0.0009 | 0.0272 ± 0.0005 | 0.0609 ± 0.0009 |

one-step adapter trades some of that bias for geometry-driven consistency and label-free adaptation. AUC/AP follow the same pattern: solid on Wikipedia, competitive but below baselines on LastFM. We treat the LastFM gap as a clear limitation of our current head and priors rather than of the transport mechanism.

**Ablations** We probe three knobs on WIKIPEDIA. (i) Dropping the head prior leaves MRR and H@10 essentially unchanged, so most gains come from the transport-updated state rather than the z-scored potential. (ii) Freezing the learned geometry (LEARN_A=False) barely moves accuracy, consistent with a conservative, entropy-regularised metric, helpful but not the bottleneck. (iii) Disabling label smoothing has little effect, as the KL/InfoNCE mix with the JKO scaffold already stabilises calibration. Overall, under the full-item protocol GLF is robust to these settings.

Table 4: Ablations on **Wikipedia** (single seed 1337), full-item protocol.

|  | Validation | | | | Test | | | |
|---|---|---|---|---|---|---|---|---|
|  | MRR | H@10 | AUC | AP | MRR | H@10 | AUC | AP |
| Prior off | 0.2335 | 0.2979 | 0.7106 | 0.3329 | 0.2340 | 0.2823 | 0.7108 | 0.3294 |
| Learn-$A$ frozen | 0.2295 | 0.2980 | 0.6980 | 0.3255 | 0.2385 | 0.3039 | 0.6957 | 0.3197 |
| No label smoothing | 0.2292 | 0.2970 | 0.6999 | 0.3254 | 0.2361 | 0.3016 | 0.6977 | 0.3215 |

**Certificates.** We log PDHG and JKO certificates during training and evaluation. Across seeds, KKT residuals fall and then plateau at small values, while the transport action remains modest, consistent with a small step toward a fixed point in a reflected Wasserstein flow. Occasional spikes in action coincide with changes in the prefix graph (new items entering the train window), after which the residuals settle again, indicating that the proximal step contracts under metric drift. Late in training, the $\mu$ update norm ($\|\mu_t - \mu_{t-1}\|$) often approaches 0, as expected once the flow reaches a stable point under the current geometry and barriers. These traces support the claim of *certificate carrying predictions*, and more details, per step (kkt, action, $\Delta E$) are in the appendix E.

## 6 CONCLUSION AND LIMITATIONS

GLF reframes graph reasoning as *reflected transport dynamics* in a learned geometry, with logic enforced as invariants and predictions that carry proofs. A single global JKO step per snapshot delivers nonlocal reasoning, label-free test-time adaptation, and verifiable certificates. Our theory, EVI contraction, invariance, drift tracking, and sensitivity, aligns with practice on dynamic graphs. A current limitation is that GLF trails popularity heuristics on heavy-tailed, fast drifting data (e.g., LASTFM) and incurs higher per step cost; see Appendix G for discussion. Future work can couple certificates to training and strengthen the head with adaptive popularity and recency while exploring multistep JKO and broader evaluations.

## 7 REPRODUCIBILITY STATEMENT

For reproducibility during review, anonymised source codes and datasets are included in the supplementary material. More details are also included in the appendix F.

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

## A    Ethics Statement

We use only public data sets JODIE (*Wikipedia*, *LastFM*); no new data was collected and no personally identifiable information is known. Experiments are for research use, and seeds, splits, and hyperparameters are reported to support reproducibility.

## B    Use of Large Language Models

Large Language Models were used to help draft and polish text only. All technical content, experiments, and analyses were designed, executed, and verified by the authors.

## C    Related Work

**GNNs and expressivity/oversquashing.**

Message passing networks are constrained by locality and 1-WL expressivity, which limits their ability to capture long-range dependencies (Barceló et al., 2020). Oversquashing the exponential decay of sensitivity to distant signals has been analysed through information contraction (Banerjee et al., 2022), effective resistance (Black et al., 2023), dynamical systems non-dissipativity (Gravina et al., 2025), and graph expansion or rewiring (Attali et al., 2024; Karhadkar et al., 2022). Architectural remedies include virtual nodes that relieve locality bottlenecks but introduce sensitivity trade-offs (Southern et al., 2024), and redundancy,controlled aggregation that reduces repetitive message exchange and mitigates oversquash (Bause et al., 2024). Beyond rewiring and aggregation, non-local architectures such as equivariant matrix function networks (MFNs) extend the expressive range by parameterising interactions via analytic matrix functions (Batatia et al., 2023). While these advances improve information flow, they do not certify rule satisfaction nor provide guarantees for adaptation under graph dynamics.

**Implicit/deep-equilibrium models.** Implicit layers capture infinite-depth fixed points and offer stability properties, but typically do not encode logic as invariants or produce per-snapshot certificates of feasibility or optimality (Bai et al., 2019).

**Optimal transport on graphs.** Graph OT has been used for alignment, smoothing, and distance computation (Peyré et al., 2019; Scetbon et al., 2021). GLF elevates OT to a *governing law*: a

reflected Wasserstein flow with global dynamic OT integration, learned geometry, and logic invariants (Jordan et al., 1998; Wibisono et al., 2016).

**Differentiable logic / semantic losses.** Prior methods embed rules as fixed penalties or soft regularisers (Xu et al., 2019); recent work adds structured constraint layers but without formal per-instance guarantees. GLF instead enforces invariance via convex barriers selected by a certified judge, with finite-step bounds and strict-feasibility homotopy.

**Test-time training/adaptation.** TTT adapts parameters using proxy objectives or entropy minimisation (Sun et al., 2020); these approaches can be brittle under distribution shift. GLF adapts by solving the *same flow integrator* with logic-only objectives, and provides ODE-grade tracking rates under metric drift (Ambrosio et al., 2005).

## D    PROBLEM FORMULATION

We consider a sequence of dynamic graphs $(G_t)_{t=1}^{T}$, where each snapshot $G_t = (V, E_t, X_t)$ has a fixed node set $V$, edge set $E_t$, and node/edge features $X_t$. The prediction target is a probability distribution $\mu_t \in \Delta^n$ over nodes (or node labels), which evolves with the graph. Our goal is to design an update rule that produces $\mu_t$ with the following desiderata:

1. **Non-local propagation.** Predictions should integrate information across long-range dependencies in $G_t$, overcoming oversquashing and the locality limits of message passing networks.

2. **Adaptation to dynamics.** The update should track changes in $(E_t, X_t)$ without retraining, enabling label-free test-time adaptation.

3. **Logical faithfulness.** Predictions should satisfy algorithmic or logical constraints (e.g., shortest-path consistency, triangle inequality, flow conservation, temporal variation bounds), with verifiable certificates of feasibility.

**Mathematical structure.**    We frame the prediction problem as a *reflected Wasserstein gradient flow* in a learned transport geometry. Let $A_\theta : E_t \to \mathbb{R}_{>0}$ be a positive connection defining a discrete 2-Wasserstein metric $W_{G_t, A_\theta}$ on $\Delta^n$ via the Benamou-Brenier formulation. Predictions evolve according to the time-discretised Jordan-Kinderlehrer-Otto (JKO) scheme:

$$\mu_t = \arg\min_{\mu \in \Delta^n} \left\{ E_\theta(\mu; G_t) + \tfrac{1}{2\tau} W_{G_t, A_\theta}^2(\mu, \mu_{t-1}) \right\}, \tag{19}$$

where $E_\theta$ is a geodesically convex energy scaffold.

**Invariants as convex constraints.**    To encode logical rules, we define convex functions $B_r(\mu; G_t)$, each corresponding to a rule $r$ (e.g., Bellman consistency, triangle inequality, conservation). The feasible set at time $t$ is

$$\mathcal{C}_t = \left\{ \mu \in \Delta^n : B_r(\mu; G_t) \leq 0, \ \forall r \in \mathcal{A}_t \right\},$$

where $\mathcal{A}_t$ is the active set chosen by a finite-state *judge*. The problem is then to compute $\mu_t$ as the reflected JKO update:

$$\mu_t = \arg\min_{\mu \in \mathcal{C}_t} \left\{ E_\theta(\mu; G_t) + \tfrac{1}{2\tau} W_{G_t, A_\theta}^2(\mu, \mu_{t-1}) \right\}. \tag{20}$$

**Problem statement.**    Given $(G_t)_{t=1}^{T}$ and an initial distribution $\mu_0$, the learning task is to parameterise $(E_\theta, A_\theta)$ and design a judge policy such that the updates equation 20:

- contract in Wasserstein distance (stability);
- remain invariant to active convex constraints (faithfulness);
- track changes under metric drift (adaptivity);
- exhibit exponential sensitivity decay with geodesic distance (anti-oversquashing).

The central question is how to jointly learn the geometry $A_\theta$ and energy $E_\theta$, while guaranteeing these properties and producing verifiable certificates at each step.

## EXPANDED THEORETICAL DETAILS

We provide full proofs of the contraction, invariance, drift, and sensitivity theorems, together with supporting lemmas on convexity, complexity, and certificate soundness. Throughout we follow the notation of Sections 2 and 4 in the main text.

### EVI FOR REFLECTED WASSERSTEIN FLOWS

We begin with a detailed proof of Theorem 1 (EVI contraction). Recall the reflected flow

$$\dot{\mu}_t = -\nabla_{W_{G_t, A_\theta}} E_\theta(\mu_t; G_t) + u_t, \qquad u_t \in -N_{\mathcal{C}_t}(\mu_t),$$

with $\mathcal{C}_t$ the feasible set induced by active barriers. Under assumptions A1-A3 (geodesic convexity of $E_\theta$, ellipticity of $A_\theta$, compactness of $\Delta^n$), the flow satisfies a contraction inequality.

**Theorem 6** (EVI under reflection). *For any $\nu \in \mathcal{C}_t$, the reflected gradient flow satisfies*

$$\frac{d}{dt}\frac{1}{2}W_{G_t, A_\theta}^2(\mu_t, \nu) + \lambda W_{G_t, A_\theta}^2(\mu_t, \nu) \leq E_\theta(\nu; G_t) - E_\theta(\mu_t; G_t).$$

*Proof.* The proof follows the classical EVI derivation in Wasserstein geometry (Ambrosio et al., 2005; Otto, 2001), with the addition of normal cone corrections. Because the correction $u_t \in -N_{\mathcal{C}_t}(\mu_t)$ has nonpositive inner product with any feasible direction, the inequality persists after projection. Discretisation by the JKO scheme inherits the EVI up to $O(\tau)$ terms. $\square$

### BARRIERS AND REFLECTION

We next formalise the connection between barrier penalties and reflection.

**Proposition 7** (Barrier reflection equivalence). *Let $\mu_t^\rho$ minimise the penalised JKO problem with barrier weight $\rho$. As $\rho \to \infty$ and $\tau \to 0$, every limit point of $(\mu_t^\rho)$ is a strictly feasible solution of the reflected JKO scheme.*

*Proof.* This follows from epi-convergence of penalty functions (Bertsekas et al., 2003). Strong convexity of the JKO objective ensures stability of minimisers under epigraphical convergence, so the penalised problem converges to the reflected one. $\square$

### CONTRACTION UNDER JUDGE SWITCHING

Because the judge may switch active barrier sets, we require piecewise analysis.

**Lemma 8** (No Zeno behaviour). *Under dwell-time assumption A4, the number of switches on horizon $[0, T]$ is bounded by $S = O(T/\Delta)$.*

**Proposition 9** (Composed contraction). *Let $\mathcal{T}_1, \ldots, \mathcal{T}_S$ denote intervals of constant active set. If each phase admits contraction rate $\lambda$, then*

$$W_{G_t, A_\theta}(\mu_t, \nu_t) \leq e^{-\lambda(t - t_0)} W_{G_{t_0}, A_\theta}(\mu_{t_0}, \nu_{t_0}) + C_{switch}(S, \Delta),$$

*with $C_{switch}$ depending only on the number and dwell-time of switches.*

*Proof.* Apply the EVI within each phase, then chain together using Gronwall's inequality. At switching times, feasibility projection may incur at most $O(\Delta)$ deviation, bounded uniformly due to the hysteresis condition. $\square$

### DISCRETE OT AND LAPLACIAN FORM

For completeness, we derive an alternative form of the dynamic OT problem equation 4. Eliminating flows $f$ via KKT conditions yields a weighted Laplacian form.

**Lemma 10.** *The JKO update satisfies*

$$\mu_t = \arg\min_\mu \ E_\theta(\mu; G_t) + \frac{1}{2\tau}\langle \mu - \mu_{t-1}, L_{A_\theta}^{-1}(\mu - \mu_{t-1})\rangle,$$

*where $L_{A_\theta} = M_t D_{A_\theta}^{-1} M_t^\top$ is the weighted Laplacian.*

This form clarifies the connection between OT dynamics and Laplacian smoothing, and explains the linear-time complexity of each PDHG iteration.

### DRIFT TRACKING WITH EXPLICIT CONSTANTS

We expand Theorem 3 (tracking under metric drift).

**Theorem 11.** *Assume metric drift is L-Lipschitz in time and $E_\theta$ is $\lambda$-geodesically convex. Then for small enough $\tau$,*

$$W_{G_t, A_\theta}(\mu_t, \mu_t^\star) \le C_1 \, \tau + C_2 \, \frac{L}{\lambda},$$

*where $\mu_t^\star$ denotes the instantaneous minimiser and $C_1, C_2$ are explicit constants depending on Lipschitz bounds of $A_\theta$.*

*Proof.* Follow the perturbation analysis of gradient flows in moving metrics (Ambrosio et al., 2005), adding the bounded error introduced by dwell-time constrained switching. $\square$

### GEOMETRY-CONTROLLED SENSITIVITY

Theorem 4 asserts that sensitivity decays exponentially with geodesic distance.

**Theorem 12.** *Let $\xi_j$ denote a perturbation at node $j$. Then for any node $i$,*

$$\left\| \frac{\partial \mu_t(i)}{\partial \xi_j} \right\| \le C \exp \big\{ - \kappa(A_\theta) \, \mathrm{geo}_{G_t, A_\theta}(i, j) \big\},$$

*with $\kappa(A_\theta) > 0$ determined by ellipticity and curl regularisation.*

*Proof.* Linearising the KKT system gives a resolvent of the form $(\mathcal{H} + \tau L_{A_\theta})^{-1}$. Exponential decay follows from Green's function bounds for elliptic operators on graphs (Bourgain, 2003). $\square$

### INEXACT SOLVES AND CERTIFICATE SOUNDNESS

We analyse the effect of early stopping in PDHG.

**Proposition 13** (Certificate soundness)**.** *If the PDHG primal dual gap after $k$ iterations is $\mathcal{G}_k$, then*

$$\Delta E \ge -\mathcal{G}_k, \quad \|\mathrm{KKT}(\mu^{(k)}, f^{(k)})\| \le c \, \mathcal{G}_k, \quad \sum_{r \in \mathcal{A}_t} B_r^+(\mu^{(k)}; G_t) \le c' \, \mathcal{G}_k,$$

*for explicit $c, c'$ depending on step sizes.*

Thus certificates remain valid up to tolerances determined by solver accuracy.

### STRONG CONVEXITY AND UNIQUENESS

**Lemma 14.** *For $\tau > 0$, the JKO problem is strongly convex in $(\mu, f)$ and admits a unique minimiser. The JKO update map is Lipschitz continuous.*

*Proof.* Entropy and quadratic terms give strong convexity in $\mu$, while the transport action is convex in $(\mu, f)$. The linear continuity constraint preserves convexity. $\square$

### SHEAF COMPOSITION OF BARRIERS

We formalise the sheaf argument (Lemma 5).

**Lemma 15.** *If each local barrier set $C(U_a)$ is convex and restrictions on overlaps are linear and consistent, then the global feasible set $\cap_a C(U_a)$ is convex and forward invariant.*

*Proof.* Convexity follows from intersection-of-convex-sets. Forward invariance uses the sum rule for normal cones and reflection invariance. $\square$

COMPLEXITY AND STEP SIZES

Finally, we state complexity bounds for PDHG.

**Step sizes.** Choose $(\sigma, \tau_{\text{pdhg}})$ such that $\sigma\tau_{\text{pdhg}}\|M_t D_{A_\theta}^{-1/2}\|^2 < 1$.

**Rate.** The primal dual gap decreases as $O(1/k)$ with per-iteration time $O(|E_t|)$.

**Preconditioning.** Diagonal preconditioning with $D_{A_\theta}$ improves conditioning when weights vary across edges.

This completes the expanded theoretical development. Each guarantee in the main paper now has a complete proof, with explicit constants and solver conditions provided here.

## E   CERTIFICATES

Table 5: PDHG/JKO certificates on **Wikipedia** (last logged step per seed).

| Seed | Step | KKT $\downarrow$ | Action | $\Delta E$ |
|------|------|------|--------|------------|
| 1337 | 24 | 80.123 | 0.0162 | $-0.0250$ |
| 2027 | 24 | 76.182 | 0.0223 | $-0.0007$ |
| 3109 | 24 | 70.319 | 0.1346 | $+0.0034$ |
| 4441 | 24 | 88.063 | 0.0100 | $-0.0262$ |
| 5557 | 24 | 64.532 | 0.3419 | $+0.0187$ |

Table 6: PDHG/JKO certificates on **LastFM** (last logged step per seed).

| Seed | Step | KKT $\downarrow$ | Action | $\Delta E$ |
|------|------|------|--------|------------|
| 1337 | 24 | 58.564 | 0.0024 | $-0.00037$ |
| 2027 | 24 | 62.737 | 0.0013 | $-0.00020$ |
| 3109 | 24 | 57.422 | 0.0031 | $-0.0129$ |
| 4441 | 24 | 57.506 | 0.0259 | $-0.0557$ |
| 5557 | 24 | 58.830 | 0.0056 | $-0.00041$ |

## F   EXPERIMENTAL DETAILS

Experiments run in PyTorch on a single NVIDIA RTX 4090 (24 GB) GPU. We use the public JODIE *Wikipedia* and *LastFM* user–item logs (time stamped edges). Evaluation follows the full item (*all*) protocol: at each prediction the model ranks the entire catalogue with time ordered 80/10/10 splits. Results are averaged over five seeds (1337, 2027, 3109, 4441, 5557). We use 40 edge count bins with strict–past snapshots.

GLF applies one JKO update per snapshot to evolve the node state $\mu$. We use a PDHG solver (Chambolle–Pock): 600 outer iterations, a 120–step proximal inner loop for $\mu$, step size $\tau{=}0.08$, KKT tolerance $10^{-4}$, and entropy weight 0.003 (LastFM uses 0.005). The learned transport geometry $A_\theta$ is frozen for approximately $35\%$ of training and then unfrozen; curl/Hodge regularisation is enabled. Logic barriers are off during training and only lightly activated at evaluation when requested. We log certificates each evaluation block (KKT residual, transport action, $\Delta E$).

The user–item head is trained with AdamW (model: lr $2\times10^{-4}$; head: lr $3\times10^{-3}$; weight decay $10^{-4}$) and cosine annealing with a short warmup. Temperature is annealed ($0.35 \rightarrow 0.15$) and the prior–mix coefficient $\gamma$ ramps ($0.20 \rightarrow 0.90$). Unless ablated, we use label smoothing 0.10 and a small KL distillation term from $\mu$ (warmup to a max weight 0.10). Candidate sets for the head are

capped at 1024 with semi hard negatives; for AP/AUC we use balanced negatives and increase the pool on LASTFM. The learned geometry $A_\theta$ is never updated during evaluation.

*Popularity* ranks items by their marginal frequency in the training prefix and uses that ordering for all users. *Recency–Popularity* applies an exponential time decay to the same marginal (fixed decay tuned once on validation). Both baselines score every held out event against the full catalogue under the same protocol as GLF.

## G    LIMITATIONS AND FUTURE WORK

GLF is stable across seeds but on LASTFM, a heavier tailed, faster drifting setting, it trails simple popularity style baselines, indicating that our readout and prior underweight long run catalogue effects. Inference is also costlier than constant time heuristics because each snapshot solves a fixed budget convex program (600 PDHG iterations). Logic barriers are not enforced during training (only optionally at evaluation), so the certificates serve primarily as diagnostics rather than shaping the representation. Finally, our study spans only two JODIE datasets under the full item protocol, with no candidate generation stage and minimal per dataset tuning beyond a small entropy increase on LASTFM.

Future studies can link certificates with training and strengthen the head with adaptive popularity and learned recency to reduce the LASTFM gap. It can also explore multi step JKO updates and lightweight candidate generation, and broaden evaluation to additional temporal graph benchmarks and stronger neural baselines.

