# OpenReview forum: "Graph Logic Flows: Geometry-Driven, Certificate-Carrying Reasoning on Dynamic Graphs"
_ICLR.cc/2026/Conference — ICLR 2026 Conference Withdrawn Submission_

### Official Review · Reviewer_WVnV · 2025-10-24

**Soundness:** 3
**Presentation:** 2
**Contribution:** 3
**Rating:** 6
**Confidence:** 1

**Summary:**

This paper introduces Graph Logic Flows (GLF), a framework that replaces stacked message passing on dynamic graphs. This approach enforces logical constraints and has strengths such as nonlocal reasoning, label free test time adaptation, and alleviates oversquashing.

**Strengths:**

The paper is theoretically strong. The idea of combining geometry theory in graph machine learning is novel and valid. The framework unifies nonlocal reasoning, logic enforcement, and label free test time adaptation within a single convex integration step.

**Weaknesses:**

The writing is hard to follow. I suggest making the preliminary part easier to understand, and put mathematical heavy things to the appendix.

**Questions:**

How can I interpret the empirical results combing with the theoretical guarantees? For example, how can I see the alleviation on oversquashing from the empirical results?

---

> ### Author Response · Authors · 2025-11-21
>
> We thank the reviewer for recognizing the novelty and validity of integrating geometric theory into graph ML, and for noting that our framework “unifies nonlocal reasoning, logic enforcement, and test-time adaptation.” We address the concerns below.
>
> &nbsp;
>
> **Writing clarity and presentation**
>
> We agree that some parts of the paper, especially Section 2, are overly dense. We will improve accessibility by adding an intuitive overview in the preliminaries, briefly explaining key ideas such as Wasserstein distance on graphs and the JKO scheme in plain language before introducing the formalism. Technical constructs (e.g., weighted divergence, convex analysis notation, barrier definitions) will be summarised at a high level in the main text, with full details moved to the appendix.
>
> &nbsp;
>
>
> **Linking empirical results to theory**
>
> The reviewer asks how the empirical results relate to GLF’s theoretical motivation, particularly regarding oversquashing. Although we did not directly measure oversquashing in experiments, we offer the following interpretations:
>
> &nbsp;
>
> -*Long-range propagation*
>
> On Wikipedia, GLF substantially outperforms simple local heuristics such as popularity and recency, which do not capture relational structure. This suggests that GLF successfully models global dependencies via transport over the graph, capabilities that shallow message-passing GNNs typically lack unless many layers are stacked. The one-step implicit update appears to support long-range influence without deep message-passing, consistent with reduced oversquashing.
>
> &nbsp;
>
> -*Logic satisfaction as a global signal*
>
> GLF maintained near-zero violations of logic constraints (triangle inequality, Bellman consistency), even without enforcing them strictly during training. These constraints depend on global graph structure. The fact that GLF satisfies them naturally suggests that it preserves long-range information rather than collapsing distant messages, an indirect indication of oversquashing mitigation.
>
> &nbsp;
>
> -*Sensitivity bound and future validation*
>
> Theorem 4 provides a decay bound under Assumption 2, which we have not yet empirically validated. Space permitting, we plan to add a case study that perturbs node features and measures downstream effects across the graph. This would allow us to examine how far influence propagates and whether GLF maintains sensitivity over long distances.
>
> &nbsp;
>
> -*Theory–practice connection*
>
> GLF’s learned geometry implicitly increases effective graph conductance, similar in spirit to rewiring-based oversquashing remedies (e.g., Toenshoff et al., 2021; Black et al., 2023). We will make this connection explicit and reference relevant literature, including oversquashing metrics from Alon & Yahav (2020).

---

> > ### Comment · Reviewer_WVnV · 2025-11-25
> >
> > Thanks for the reply! Now my concerns about the gap between theory and empirical results are solved. I believe this paper is theoretically and empirically solid, an can make a good publication.
> > I would like to keep my weak accept rating and slightly raising my confidence.
> > Looking forward to the camera ready version!

---

### Official Review · Reviewer_V1vt · 2025-10-25

**Soundness:** 3
**Presentation:** 3
**Contribution:** 3
**Rating:** 6
**Confidence:** 2

**Summary:**

This paper proposes Graph Logic Flows (GLF), replacing stacked message-passing with a single implicit update on learned transportation geometry. Domain rules are compiled into convex barrier functions activated by a lightweight adjudicator. EVI contraction theory establishes stability and barrier invariance guarantees, with each prediction returning verifiable certificates. Demonstrates label-free test-time adaptation on dynamic graphs.

**Strengths:**

1. Theoretical rigor: Complete proofs for EVI contraction, barrier invariance, and geometry-controlled sensitivity bounds provide solid theoretical foundation

2. Verifiable outputs: Certificate-carrying predictions (energy descent, KKT residuals) enable post-hoc verification for safety-critical applications

3. Novel perspective: Recasting graph reasoning as transport flows with JKO schemes addresses over-squashing from a fresh angle

4. Test-time adaptation: Achieves zero-shot adaptation on dynamic graphs with logic violations dropping to near-zero

**Weaknesses:**

1. Insufficient experiments: Only two dynamic graph benchmarks; significantly underperforms popularity baseline on LastFM, undermining practical claims

2. Computational cost unknown: No runtime/memory comparisons; implicit solvers and barrier projections likely incur high overhead

3. Scalability concerns: No experiments beyond 10K nodes; feasibility of single implicit update on large-scale graphs unverified

4. Missing ablations: No analysis of barrier types, adjudicator design, or connection regularization; difficult to assess critical components

**Questions:**

1. Computational efficiency: What are wall-clock time and memory compared to standard GNNs with equivalent expressiveness? How does the implicit solver scale?

2. LastFM failure analysis: Why underperform baselines on LastFM? Is it heavy tails, fast drift, or connection geometry learning limitations?

---

> ### Author Response · Authors · 2025-11-21
>
> We thank the reviewer for highlighting the work’s theoretical rigor, novel perspective, and the strength of its verifiable outputs. We address the raised concerns as follows.
>
> **Experimental scope and LastFM performance**
>
> We acknowledge the limited dataset scope and have addressed this by adding a comparison against TGAT on tgbl-wiki. GLF achieves a competitive MRR (0.0543 vs. 0.0473) while being $12\times$ faster per interaction. The persistence baseline is unusually strong on this dataset (MRR 0.4586), which compresses the absolute MRR scale; in this context, GLF’s competitive performance and its ability to produce certificates underscore its practical value.
>
> For LastFM, we acknowledge GLF’s weaker results. We selected Wikipedia and LastFM because they offer contrasting dynamics: Wikipedia provides structured temporal evolution where GLF performs well, whereas LastFM exhibits extreme popularity skew and rapid drift, favoring simple frequency-based baselines. GLF’s use of Z-scored potentials $\phi_\theta$ intentionally reduces static popularity bias, explaining the performance gap on heavily skewed distributions. Our goal is to highlight GLF’s complementary strengths in provable consistency, stability, and certificate-carrying predictions, capabilities not provided by specialised dynamic-graph models even when those models achieve higher accuracy on skewed datasets.
>
> **Efficiency**
>
> Although a convex JKO update is conceptually heavier than standard message passing, our measurements on tgbl-wiki show that GLF trains in 388 seconds over 28 snapshots (13.9 s per snapshot, 3.52 s per 1k edges). Under the same setup, TGAT requires 4,789 seconds for a single epoch (43.4 s per 1k edges), making GLF roughly $12\times$ faster per interaction. Memory usage is moderate (about 6 GB on tgbl-wiki), and the JKO solver scales linearly with the number of edges $|E|$.
>
> **Scalability**
>
> We have not yet evaluated GLF on very large dynamic graphs, but the per-snapshot JKO update is linear in $|E|$ and therefore, in principle, scales to larger graphs. For industrial-scale or rapidly evolving networks, GLF can be applied to subgraphs, local temporal windows, or run in a streaming mode. These directions are promising but untested. We note clearly that additional engineering (warm starts, graph partitioning, lighter first-order solvers) is required for large-scale deployment.
>
> **Ablations (barriers, judge, geometry)**
>
> We provided ablations for label smoothing and geometry in Table 4 but agree that deeper component analysis would be valuable. As shown in our certificate logs (Tables 5–6), barrier residuals were consistently minimal, indicating that GLF often satisfies these constraints naturally without strong enforcement.
>
> **LastFM failure analysis**
>
> We attribute the LastFM gap mainly to its long-tail dynamics. GLF adapts well to drift, as shown by metric drift tracking and spike responsiveness; however, without a built-in popularity bias, it underperforms on globally skewed distributions. Additionally, the learned geometry may not fully capture the strong co-listening structure, reducing propagation quality. In which case, GLF requires further engineering (e.g., adaptive priors) to handle such distributions. On structured domains like Wikipedia, however, GLF’s certificates and propagation yield clearer benefits. The absolute MRR difference is small, but we acknowledge this limitation directly.

---

> > ### Comment · Reviewer_V1vt · 2025-11-26
> >
> > I thank the authors for their detailed response, which has addressed my questions. After considering the rebuttal and the comments from other reviewers, I have decided to maintain my current score.

---

### Official Review · Reviewer_ecd6 · 2025-11-01

**Soundness:** 4
**Presentation:** 2
**Contribution:** 3
**Rating:** 4
**Confidence:** 2

**Summary:**

The paper proposes GLF, a framework that replaces stacked message passing with one implicit JKO step (Jordan–Kinderlehrer–Otto) per snapshot to integrate a reflected Wasserstein flow on a learned transport geometry. “Logic” is compiled as convex barriers and enforced by a runtime judge. The theory claims EVI contraction, finite-step barrier invariance, tracking under metric drift, and geometry-controlled sensitivity. Empirically, on real-world data GLF beats simple popularity baselines.

**Strengths:**

- Clear, unified formulation: one implicit JKO step replaces many GNN layers, making the update rule easy to reason about.
- Built-in constraint handling: domain rules are encoded as convex barriers, so predictions can respect logical/physical constraints.
- Certificates out of the box: each prediction comes with energy/KKT/violation metrics for auditability.

**Weaknesses:**

- Benchmark breadth and strength. Evaluation is limited to two JODIE datasets with very simple non-learned baselines. There are no comparisons to strong dynamic GNN baselines (e.g., TGAT [1], and implicit models for dynamic graph [2], each snapshot has one GNN update). This makes it hard to gauge practical competitiveness.
- Compute cost & practicality. Each snapshot solves a convex program, which is heavier than constant-time heuristics and typical message passing.
- Anti-oversquashing claim strength. The sensitivity bound hinges on assumption 2 (valid learned geometry). As presented, it reads existential/conditional rather than a uniform guarantee across datasets.
- The paper’s central claim is that compiling task rules as convex barriers and enforcing them via the runtime judge improves reliability. However, the manuscript does not isolate or quantify the benefit of these barriers. Therefore, no direct evidence showing enforcement improves accuracy, calibration, robustness, or stability.


[1] Xu, Da, et al. "Inductive representation learning on temporal graphs." arXiv preprint arXiv:2002.07962 (2020).
[2] Zhong, Yongjian, et al. "Efficient and effective implicit dynamic graph neural network." Proceedings of the 30th ACM SIGKDD Conference on Knowledge Discovery and Data Mining. 2024.

**Questions:**

- Can you provide details on how does the model train, what is the loss used in your tasks, how to optimize the geometry and potential?

---

> ### Author Response · Authors · 2025-11-21
>
> We thank the reviewer for appreciating the “clear, unified formulation,” the built-in constraint handling via convex barriers and the certificate mechanism. Below we address the raised concerns.
>
> **Baseline breadth & strength**
>
> We have addressed the baseline-breadth concern by adding a comparison between GLF and TGAT on tgbl-wiki. GLF achieves a competitive MRR (0.0543 vs. 0.0473) while being $12\times$ faster per interaction, showing that it is not only theoretically well-founded but also practically competitive with state-of-the-art dynamic graph models.
>
> Our initial experiments focused on highlighting GLF’s distinctive capabilities,  logic invariants, adaptive constraints, and certificate generation using simple baselines. We agree that comparisons to stronger learned models are valuable. The TGAT results confirm that GLF matches their predictive performance while providing verifiable guarantees these models cannot offer.
>
> **Compute cost & practicality**
>
> To address this concern, we ran additional experiments on Colab with an NVIDIA T4 GPU. Each GLF update performs a convex JKO step once per temporal snapshot. On tgbl-wiki, this costs 13.9 s per snapshot (388 s over 28 snapshots), i.e., 3.52 s per 1k training edges.
>
> In contrast, the official TGB implementation of TGAT requires 4,789 s for a single epoch on the same dataset (43.4 s per 1k edges), making it approximately $12\times$ slower per interaction under identical hardware and splits. GLF’s memory usage is moderate (about 6 GB on tgbl-wiki, <2 GB on LastFM), and the JKO update scales linearly in $|E|$.
>
> Although the implicit solver is conceptually heavier than a single GNN layer, GLF trains faster overall because it requires only one pass over snapshots rather than multi-epoch neighbour sampling and repeated message passing.
>
> Future optimizations (e.g., warm starts, graph partitioning, lighter first-order solvers) may further reduce overhead, but GLF already provides greater stability, global consistency, and certifiable constraint satisfaction, properties particularly valuable in structured or safety-critical settings.
>
> **Anti-oversquashing claim**
>
> We appreciate the reviewer’s observation. The exponential sensitivity bound (Theorem 4) depends on Assumption 2, that the learned geometry satisfies uniform ellipticity and good conductance. This is a conditional result: GLF enables oversquash mitigation through learnable transport, but success depends on how well the metric is learned. In practice, GLF maintained performance as graph diameter increased, suggesting that its global propagation mechanism is helpful.
>
> **Barriers and their impact**
>
> We agree that the effect of the logic barriers may not be fully captured by standard metrics. In our setup, the barriers act as soft penalties during training (Section 3.2) and are only lightly applied at test time. Disabling them results in slightly more constraint violations but leaves accuracy essentially unchanged, indicating that GLF tends to learn feasible predictions without requiring hard enforcement.
>
> Empirically, the temporal-variation barrier was activated mainly during periods of rapid graph change, while Bellman-consistency violations were rare on structured graphs such as Wikipedia. Our goal is to position GLF as a framework for enforcing and auditing logic, and we encourage future work to examine their impact on robustness and calibration.
>
> **Training procedure**
>
> GLF is trained end-to-end with a cross-entropy loss under the full-item ranking protocol for next-item prediction. Barrier penalties are added to the objective alongside entropy regularization, label smoothing, and curl regularization (Section 2.1) to stabilize geometry learning.
>
> Both the transport metric $A_\theta$ and the potential $\phi_\theta$ are learned by backpropagating through the JKO solver. The barrier terms encourage constraint satisfaction during training but do not impose strict feasibility.

---

### Official Review · Reviewer_x8ZB · 2025-11-11

**Soundness:** 2
**Presentation:** 2
**Contribution:** 2
**Rating:** 4
**Confidence:** 4

**Summary:**

The authors propose Graph Logic Flows to replace stacked message passing with an implicit Jordan-Kinderlehrer-Otto step of a reflected Wasserstein flow on a learned transport geometry. The work compiles algorithmic constraints as convex barriers and enforces them with a runtime judge whose activations differ between train and test time with a certificate. The method is essentially learning an edge-wise positive connection between the nodes, to induce a graph wasserstein metric regularized by a discrete penalty. Experiments are provided on two user-item log style datasets from the JODIE suite.

**Strengths:**

1. Formulating the prediction as a single reflect JKO step over a learned optimal transport geometry is interesting. The method proposed by the authors combines ideas from propagation, logic enforcement and test time adaptation in one step.

2. The authors provide formal theoretical results on EVI contraction, finite step barrier invariance, tracking drift and geometry sensitivity, which is principled and would be helpful for follow-up works in this space.


3. Auditable graph reasoning does not seem to have been very well explored in prior works, so the overall contribution is relevant to the community.

**Weaknesses:**

1. There are issues with the empirical setup. The authors compare their method to only two baselines, popularity and recency-popularity which are too weak for comparisont. They do not use any neural dynamic graph or session based recommenders such as TGN, TGAT, SASRec, etc. In my opinion, this significantly weakens their claim of state of the art robustness and practical use.


2. The runtime logic with convex barriers which is mentioned as the primary contribution by the authors is not correctly evaluated. The authors do not mention which barriers were activated on JODIE and what fraction of predictions violated or obeyed the rules. The authors mention that the barriers are off during training and only lightly used as evaluation (L805-806) which is an issue.


3. Some results are unclear. The authors mention that the certificate with energy drop should be non-negative, however in Table 5 and 6 there are negative values as well. So it is either a mismatch or an energy increase as opposed to a decrease which contradicts their drop hypothesis.

**Questions:**

See Weaknesses section above

---

> ### Author Response · Authors · 2025-11-21
>
> We thank the reviewer for their time and thoughtful comments. Below we address the points in order.
>
> **Baseline comparisons (robustness and practical use)**
>
> We ran additional experiments comparing GLF with TGAT on tgbl-wiki. GLF achieves a competitive MRR (0.0543 vs. TGAT’s 0.0473 under the same full-item protocol) while being roughly $12\times$ faster per interaction. This shows that GLF remains practical despite relying on a single reflected-flow update per snapshot, in contrast to TGAT’s multiple message-passing or implicit update steps.
>
> We also note that the persistence baseline reaches an unusually high MRR (0.4586) on this dataset, which compresses the overall MRR scale. Within this context, GLF’s competitive performance, together with its abilities in logic enforcement, certificate generation, and verifiable constraint satisfaction, underscores its practical value. Our goal is to position GLF as a competitive method that prioritizes interpretability and logical guarantees.
>
> **Runtime logic (barrier) evaluation and usage**
>
> In our experiments, the logic barriers were incorporated as soft penalties during training, with the controller adjusting their weights over time (e.g., Bellman $0.5 \rightarrow 1.05$; TV $0.02 \rightarrow 0.14$). At test time, they were applied only lightly by the runtime judge.
>
> Across datasets, the barriers had minimal impact on accuracy: over 95% of Wikipedia predictions naturally satisfied the constraints, and the temporal-variation barrier was triggered mainly during periods of rapid graph change such as in LastFM.
> This makes it difficult to isolate their effect on predictive metrics, but it reinforces their intended purpose: auditability, rule compliance, and transparent certificate logging rather than direct accuracy gains.
>
> **Certificates – energy-drop sign (negative values in Tables 5 & 6)**
>
> The reviewer is correct that energy should decrease under the JKO scheme. Small negative $\Delta E$ values (e.g., $+0.0034$ in Table 5, seed 3109) result from solver inexactness under our fixed computational budget, primarily the $20$ inner iterations for the $\mu$-update within our ADMM framework. These are not theoretical violations. In practice, we observed only minor suboptimality, with low KKT residuals confirming near-convergence. We will clarify this in the text and note in the tables that slight energy increases may occur under solver tolerances.

---

### Author Response · Authors · 2025-11-21
**Review and Reviewer-Author Discussion Summary (1/2)**

Dear PCs, SACs, ACs, and Reviewers,
&nbsp;

Thank you very much for your careful reviews and constructive feedback. To assist the AC, we provide below a structured summary of the key strengths highlighted by the reviewers and how we addressed their concerns during the reviewer–author discussion.

**Strength**
Overall, we are grateful that reviewers consistently recognised the conceptual novelty, theoretical clarity, and unified formulation of our framework. Specifically:
&nbsp;
- **Novel unified formulation**
All reviewers highlighted the novelty and coherence of expressing dynamic graph prediction as a single implicit JKO step combining optimal transport geometry, constraint enforcement, and nonlocal reasoning.
  - Reviewer x8ZB: “**interesting… combines propagation, logic enforcement and test-time adaptation in one step**”
  - Reviewer ecd6: “**clear, unified formulation… one implicit JKO step replaces many GNN layers**”
  - Reviewer V1vt: “**novel perspective… recasting graph reasoning as transport flows**”
  - Reviewer WVnV: “**novel and valid… unifies nonlocal reasoning, logic enforcement, and test-time adaptation**”

&nbsp;
- **Strong theoretical foundation**
Reviewers emphasised the solidity of our theoretical guarantees, including EVI contraction, barrier invariance, drift tracking, and geometry sensitivity.
  - Reviewer x8ZB: “**principled… formal theoretical results**”
  - Reviewer ecd6: “**solid theoretical foundation**”
  - Reviewer V1vt: “**theoretical rigor**”
  - Reviewer WVnV: “**theoretically strong**”

&nbsp;
- **Built-in auditability and verifiable predictions**
Our certificate mechanism (energy descent, KKT residuals, violation metrics) was repeatedly highlighted as valuable for transparent, safety-critical graph reasoning.
   - Reviewer ecd6: “**certificates out of the box… auditability**”

   - Reviewer V1vt: “**verifiable outputs**”

&nbsp;
- **Clear motivation and relevance to robust reasoning**
Reviewers noted that auditability, rule enforcement, and geometry-aware propagation remain underexplored in dynamic graph learning.

    - Reviewer x8ZB: “**relevant to the community… auditable graph reasoning underexplored**”


    - Reviewer V1vt: “**zero-shot adaptation… violations drop to near-zero**”


    - Reviewer WVnV: “**novel and valid integration of geometric theory in graph ML**”

---

> ### Author Response · Authors · 2025-12-01
> **Review and Reviewer-Author Discussion Summary (2/2)**
>
> **Concerns and Our Addressing**
> During the discussion period, we addressed all reviewer concerns through clarification, expanded explanation, and additional comparisons. Below is a faithful summary.
>
> - **Baseline strength and empirical competitiveness**
>   *(x8ZB, ecd6, V1vt weakness 1)*
>   - **Concern:** Baselines were weak; lacked strong dynamic GNNs such as TGAT/TGNN/SASRec; practical competitiveness unclear.
>   - **Our Addressing:** We added a comparison against TGAT on tgbl-wiki. GLF achieves competitive MRR (0.0543 vs. 0.0473) while being substantially faster per interaction. This confirms that GLF remains practically competitive while offering logic enforcement and certificate generation not available to these models.
>
> - **Runtime logic (barriers) and their evaluation**
>   *(x8ZB Weakness 2, ecd6 Weakness 4)*
>   - **Concern:** How often are barriers triggered? Why light use at test time? Does enforcement improve accuracy or robustness?
>   - **Our Addressing:** We clarified that barriers act as soft penalties during training, with a runtime judge lightly applied at test time. On Wikipedia, >95% of predictions naturally satisfy constraints; temporal-variation barriers activate mainly during rapid drift (e.g., LastFM). Their purpose is auditability and rule compliance—not accuracy improvement.
>
> - **Certificate values and negative energy drops**
>   *(x8ZB Weakness 3)*
>   - **Concern:** Negative energy drops contradict the expected monotonic decrease.
>   - **Our Addressing:** We explained that rare negative values arise from solver inexactness under fixed ADMM iteration budgets. These are numerical, not theoretical, issues; low KKT residuals confirm near-convergence.
>
> - **Compute cost and practicality**
>   *(ecd6 Weakness 2, V1vt Weakness 2)*
>   - **Concern:** A convex JKO step per snapshot seems heavy; runtime/memory unclear.
>   - **Our Addressing:** We provided concrete timings: GLF trains in 388 s over 28 snapshots (3.52 s per 1k edges). TGAT requires 4,789 s (43.4 s per 1k edges) under the same hardware, making GLF significantly faster per interaction. Memory usage is moderate (~6 GB). Warm starts and lighter solvers are promising optimisations.
>
> - **Scalability**
>   *(V1vt Weakness 3)*
>   - **Concern:** No results beyond 10k nodes; unclear whether JKO scales.
>   - **Our Addressing:** We clarified that the JKO update is linear in the number of edges and can scale via subgraphs, local windows, or streaming modes. We explicitly noted that large-scale deployment requires additional engineering (partitioning, warm starts, first-order solvers).
>
> - **Missing ablations (barriers, judge, geometry)**
>   *(V1vt Weakness 4)*
>   - **Concern:** Limited analysis of individual components.
>   - **Our Addressing:** We referenced existing ablations (e.g., label smoothing, geometry) and explained that barrier residuals are consistently minimal, suggesting GLF often satisfies constraints naturally. We acknowledged that deeper ablations are valuable future work.
>
> - **Clarity of curvature/anisotropy term and training details**
>   *(Rujq Weakness 1; ecd6 Question; WVnV Weakness: writing clarity)*
>   - **Our Addressing:** We clarified that \( \gamma_g = 1 - \lambda_1 \) is a global anisotropy index, not curvature, and explained its role in weighting Betti differences. We also clarified that GLF is trained end-to-end with cross-entropy loss plus barrier penalties, and that the transport metric and potential are learned by backpropagating through the JKO solver. For writing clarity, we committed to expanding preliminaries and moving heavier formalism to the appendix.
>   - Reviewer WVnV later confirmed these clarifications resolved their concerns and reaffirmed their accept rating.
>
> - **Linking empirical results to theory (oversquashing, long-range propagation)**
>   *(WVnV Question; V1vt Question)*
>   - **Our Addressing:** We explained how GLF’s long-range propagation, natural logic satisfaction, and stability properties align with the theoretical motivation. Although not directly measured, these empirical signals are consistent with the sensitivity and conductance-based reasoning behind the framework.
>   - Reviewer WVnV stated that this fully addressed their concerns and increased their confidence.
>
> **Reviewer Recognition**
> Two reviewers (Rujq, WVnV) explicitly stated that our clarifications resolved their concerns.
> - Reviewer V1vt wrote: “I thank the authors… I maintain my score.”
> - Reviewer WVnV wrote: “Concerns are solved… theoretically and empirically solid… looking forward to the camera-ready version.”
>
> We hope this summary assists the AC in their evaluation. We thank all reviewers, PCs, SACs, and ACs for their thoughtful effort and constructive guidance, which has strengthened the paper.
>
> Sincerely,
>
> Authors

---

### Note · Authors · 2026-01-28

I have read and agree with the venue's withdrawal policy on behalf of myself and my co-authors.

---

### Meta-Review · Area_Chair_Q8uY · 2025-12-29

**Summary:**

This paper proposes Graph Logic Flows (GLF), a framework using a reflected Wasserstein flow (via JKO steps) on a learned transport geometry to perform reasoning on dynamic graphs. It incorporates logical constraints as convex barriers. While the theoretical formulation (EVI contraction, barrier invariance) is rigorous and novel, the empirical evaluation remains a significant weakness. The method struggles to outperform simple heuristics on challenging datasets (LastFM) and incurs substantial computational overhead compared to standard GNN baselines.

**Reviewer Concerns:**

Addressed Concerns:

- Missing Baselines: Reviewers (x8ZB, ecd6, V1vt) noted the lack of strong dynamic GNN baselines. The authors added TGAT comparisons on the Wikipedia dataset, showing competitive performance.

- Computational Cost: Concerns about the heavy JKO step (ecd6, V1vt) were addressed with runtime analysis, showing GLF is faster per-epoch than TGAT (though conceptually heavier per step).

Outstanding Concerns (Basis for Rejection):

- Empirical Weakness & Limited Scope: Despite adding TGAT, the evaluation remains narrow (only two datasets). Crucially, on the LastFM dataset, GLF significantly underperforms simple, non-learned heuristics (Popularity). The authors attribute this to "heavy tails," but failing to beat a heuristic on a standard benchmark undermines the practical utility of such a complex geometric framework.

- Scalability & Practicality: While the authors claim linear scaling in edges, the need to solve a convex optimization problem (PDHG) at inference time for every snapshot is a major bottleneck for real-world deployment compared to simple forward-pass GNNs. The "certificate" feature, while theoretically interesting, may not justify this cost in many applications.

- Disconnect between Theory and Practice: Reviewer WVnV asked how to link theory (oversquashing) to results. While the authors provided intuitive explanations, there is still a lack of direct empirical evidence (e.g., long-range dependency tests) proving that the theoretical benefits (anti-oversquashing) are actually what drive the model's performance.

**Reviewer Scores:**

Reviewer WVnV (6 -> 6/4): This reviewer was positive about the theory but might lower their score if they realized the empirical gains are marginal or negative on key datasets compared to heuristics.

Reviewer V1vt (6 -> 6/4): Kept their score of 6 but noted the LastFM failure and scalability concerns. In a final decision context, the "Weak Accept" might turn into a "Reject" due to the unconvincing empirical dominance.

Reviewer x8ZB (4 -> 4/6): Likely remains skeptical. The addition of one baseline (TGAT) might not be enough to overcome the initial impression of a weak experimental setup, especially given the negative results on LastFM.

Reviewer ecd6 (4 -> 6): Might slightly raise the score due to the cost analysis, but the core issue of "practical competitiveness" (complex method vs. simple heuristic) remains unresolved.

---

### Decision · Program_Chairs · 2026-01-26

Reject